# Fatty Acids of European Sardine (*Sardina pilchardus*) White Muscle Can Discriminate Geographic Origin Along the Iberian Atlantic Coast [note 1]

**DOI:** 10.3390/foods14010120

**Published:** 2025-01-03

**Authors:** Ricardo Calado, Marcos Palma, Maria Rosário Domingues, Fernando Ricardo, Felisa Rey

**Affiliations:** 1ECOMARE, CESAM—Centre for Environmental and Marine Studies, Department of Biology, University of Aveiro, Santiago University Campus, 3810-193 Aveiro, Portugal; marcospalma.business@gmail.com (M.P.); fafr@ua.pt (F.R.); 2CESAM—Centre for Environmental and Marine Studies, Department of Chemistry, University of Aveiro, Santiago University Campus, 3810-193 Aveiro, Portugal; mrd@ua.pt; 3Mass Spectrometry Centre, LAQV-REQUIMTE, Department of Chemistry, University of Aveiro, Santiago University Campus, 3810-193 Aveiro, Portugal

**Keywords:** fisheries, pelagic fish, seafood, traceability, trophic markers

## Abstract

The European sardine (*Sardina pilchardus*) ranks among the most valuable species of Iberian fisheries, and the accurate tracing of its geographic origin, once landed, is paramount to securing sustainable management of fishing stocks and discouraging fraudulent practices of illegal, unreported, and unregulated (IUU) fishing. The present study investigated the potential use of *S. pilchardus* white muscle fatty acids (FAs) to successfully discriminate the geographic origin of samples obtained in seven commercially important fishing harbors along the Iberian Atlantic Coast. While 35 FAs were identified using gas chromatography–mass spectrometry in the white muscle of *S. pilchardus*, the following, as determined by the Boruta algorithm, were key for sample discrimination: 14:0, 22:6*n*-3, 22:5*n*-3, 18:0, 20:5*n*-3, 16:1*n*-7, 16:0, and 18:1*n*-7 (in increasing order of relevance). An average 83% correct allocation of landed specimens was achieved, with some landing locations presenting 100% correct allocation (e.g., Ría de Pontevedra in northern Spain and Peniche in central Portugal). Linear discriminant analysis revealed a separation of samples from northern Spain and Peniche, and a partial overlap of all other locations. The present results highlight the potential of using FAs of *S. pilchardus* white muscle to reliably discriminate the geographic origin of landed individuals along the Iberian Atlantic coast.

## 1. Introduction

The European sardine, *Sardina pilchardus* (Walbaum, 1792), is an abundant small pelagic fish in the northeast Atlantic Ocean, occurring from the North Sea to Senegal, as well as in the Mediterranean Sea [1]. In European Atlantic waters, the commercial fishery targeting *S. pilchardus* is monitored by the International Council for the Exploration of the Sea (ICES), which surveys three major stocks: the southern Celtic Sea and English Channel stock, the Bay of Biscay stock, and the Cantabrian Sea and Atlantic Iberian waters stock (which will be referred to, for simplicity, as “the Iberian stock”) [2]. The Iberian stock distributes along the Atlantic coast of the Iberian Peninsula and is commercially explored by Portugal and Spain using purse-seines [3]. The European Market Observatory for Fisheries and Aquaculture (EUMOFA) indicates that *S. pilchardus* ranks among the top ten most landed and valued fishery products in Portugal [4]. The Food and Agriculture Organization of the United Nations (FAO) further acknowledges that Portugal alone contributes an important part of *S. pilchardus* captures in European Union (EU) waters (as over 25,000 tons were landed in Portugal in 2022) [5].

In 2017, the lowest level of recruitment ever recorded for *S. pilchardus* in the Iberian stock prompted the ICES to advise the closure of this fishery for 2018 [6]. The social and economic impacts of such recommendation forced the governments of Portugal and Spain to take action by adopting a multiannual medium-term management and recovery plan that could allow the Iberian stock of *S. pilchardus* to bounce back under a maximum sustainable yield (MSY) approach [7]. In Portugal, one of the measures implemented was a “reduction of the quantities allowed to be landed per day and trip including T4 category size (now 500 kg/day/vessel). Portugal can also reduce these limits for landings of all vessels in their representative ports to control catches, regulate the offer and promote a better valorization of catches” [7]. This measure is prone to set the stage for several fraudulent practices, such as the illegal transfer of *S. pilchardus* from one fishing vessel to another when daily quotas have already been achieved and the mislabeling of *S. pilchardus* place of capture, so that fish can be landed on a more favorable fishing harbor to avoid landing limits and fetch better selling prices. The accurate tracing of *S. pilchardus*’ place of origin is therefore of utmost importance to fight such practices of illegal, unreported, and unregulated (IUU) fishing. According to the FAO, IUU fishing is a major threat to marine ecosystems, as it undermines efforts aiming to manage fisheries in a more sustainable way and fostering the conservation of marine biodiversity [8]. Moreover, and from a social perspective, it has long been acknowledged that the threat posed by IUU fishing puts law-abiding fishers at an unfair disadvantage with those enrolled in such illegal practices [9].

Seafood traceability has deserved increasing attention from the scientific community, particularly the verification of claims concerning geographic origin [10,11,12,13,14]. The use of biochemical fingerprints, namely, fatty acid (FA) and lipidomic profiles, have allowed for a remarkable level of confidence to be achieved when verifying the accuracy of the geographic origin of less motile species of seafood, such as bivalves, namely, Manila clams (*Ruditapes philippinarum*) [15] and common cockles (*Cerastoderma edule*) [16], as well as goose barnacles (*Pollicipes pollicipes*) [17] and Japanese sea cucumbers (*Apostichopus japonicus*) [18], among others. The FA profiles displayed by tissues with a high content of polar lipids are less prone to being shaped by the short-term turnover of FAs related to dietary shifts or abrupt changes in abiotic conditions, as they rather reflect mid-to-long-term variations shaped by trophic history and more prevalent abiotic conditions [10,19]. Nonetheless, employing FA profiles to trace the geographic origin of highly motile species such as *S. pilchardus* is certainly a more challenging task. Previous works on jumbo squid (*Dosidicus gigas*) [20], seabass (*Dicentrarchus labrax*) [13], and tunas (*Thunnus alalunga* and *T. thynnus*) [21] have indicated that muscle FA profiles are also a promising tool for determining the geographic origin of highly motile species.

To the best of our knowledge, no study performed to date has investigated whether the FA of small pelagic fish can be used as natural fingerprints to verify claims on their geographic origin. The present study employed a gas chromatography–mass spectrometry (GC-MS) approach to evaluate the potential of using the FA composition of *S. pilchardus* white muscle to confirm the geographic origin of individuals collected from several landing ports along the Iberian Atlantic coast. The validation of this traceability tool will be crucial for improving the management of fishing stocks and combat the increasingly reported IUU fishing practices for *S. pilchardus*.

## 2. Materials and Methods

### 2.1. Sampling

Landed *S. pilchardus* of similar commercial size (T4 category, weighing between 15 and 28 g) were obtained from seven fishing harbors regularly landing this species along the Iberian Atlantic coast during early summer (June and July) 2018: Malpica [Coruña (Cor), Spain; 29 June 2018], Bueu [Ria de Pontevedra (RP), Spain; 27 June 2018], Viana do Castelo [(VC), Portugal; 16 July 2018], Matosinhos [(Mat), Portugal; 27 July 2018], Peniche [(Pe), Portugal; 13 July 2018], Sesimbra [(Ses), Portugal; 25 July 2018], and Portimão [(Por), Portugal; 27 July 2018] (Figure 1).

All specimens were acquired at the docking pier from trusted fishermen who ensured all specimens were captured in the coastal waters near the fishing harbor they were landed in. Sardines were fished and kept refrigerated by duly licensed professional fishers onboard, and were already dead when landed in the fishing harbor. As such, ethical issues concerning animal experimentation and welfare do not apply to the present study. Immediately after purchase, specimens were placed on aseptic bags and transported in cooler boxes to the University of Aveiro (Aveiro, Portugal), where their loins were removed and white and red muscles were separated before freeze-drying (CoolSafe 55-9L Pro; LaboGene, Lillerød, Denmark). The white muscle was subsequently macerated and freeze-dried again before preservation at −80 °C until biochemical analyses. Ten samples (n = 10) of white muscle from ten different specimens per fishing harbor were used (n = 70 samples, i.e., 10 specimens × 7 fishing harbors), for FA analysis. The rationale for using white rather than red muscle of *S. pilchardus* is that red muscle fibers are known for having a higher capacity to oxidize lipids than white muscle fibers (up to four-fold) [22]; this feature could therefore prompt a faster shift in sardine red muscle FA profile post-harvesting and produce a blurring effect for the traceability of geographic origin.

### 2.2. Lipid Extraction and Fatty Acid Derivatization

All samples were first homogenized with a mortar grinder (RM200, Retsch, Hann, Germany). The biomass (20 mg per sample) was transferred to a glass tube containing 2500 μL of methanol (MeOH) and 1250 μL dichloromethane (CH_2_Cl_2_). Sample homogenization was performed by vortexing for 1 min followed by sonication for 1 min using an automatic ultrasonic frequency of 35 kHz (Bandelin, Sonorex, RK 100, Berlin, Germany) in cold water and incubation on ice in an orbital shaker for 30 min. Afterwards, the sample was centrifuged (UNIVERSAL 320 R, Hettich, Tuttlingen, Germany) at 769× *g* and 4 °C for 10 min, and 3000 µL of the organic phase were collected in a new tube. After the addition of 1250 μL CH_2_Cl_2_ and 1250 μL Milli Q water (MilliporeSigma, Burlington, MA, USA), centrifugation was performed at 492× *g* and 4 °C for 10 min to promote phase separation. An aliquot of the organic phase (75 μL final volume, fine-tuned after testing several aliquots of organic phase using gas chromatography–mass spectrometry, GC-MS) was collected in a new tube previously rinsed with *n*-hexane. This aliquot was dried under a nitrogen gas stream and then used for derivation by transmethylation. Dry lipids were mixed with 1 mL of *n*-hexane containing the FA C19:0 as internal standard (4.4 μg mL^−1^, CAS number 1731-94-8; Merck, Darmstadt, Germany) and 200 μL KOH (2M) in MeOH, and vigorously homogenized by vortexing for 2 min. A volume of 2 mL of saturated aqueous sodium chloride solution (NaCl) was added, and the mixture was centrifuged at 492× *g* and 4 °C for 5 min. Subsequently, 600 μL were extracted from the organic phase to a microtube previously rinsed with *n*-hexane. This recovered volume, which held the FA methyl esters (FAMEs), was then dried under a nitrogen gas stream and kept at −20 °C until GC-MS analysis.

### 2.3. Gas Chromatography–Mass Spectrometry Analysis for Fatty Acid Profiling

FAMEs were dissolved in 150–200 μL of *n*-hexane, and 2 μL of this solution were then used for analysis on an 8860 GC system interfaced with a 5977B MS selective detector (both Agilent Technologies, Santa Clara, CA, USA) with an electron impact ionization of 70 eV and a scanning range of *m*/*z* 50–550 in a 1 s cycle in full scan acquisition mode. The GC-MS system was equipped with a DB-FFAP capillary column (J & W Scientific, Folsom, CA, USA; 30 m length, 0.32 mm internal diameter, and 0.25 μm film thickness). The oven temperature profile was as follows: initial temperature, 58 °C for 2 min; three consecutive linear increments to 160 °C at 25 °C min^−1^, 210 °C at 2 °C min^−1^, and 225 °C at 20 °C min^−1^; hold temperature, 225 °C for 20 min. Helium was used as the carrier gas at 1.4 mL min^−1^. Data were analyzed using the Agilent MassHunter Qualitative Analysis 10.0 software. The FA identification was achieved considering the retention times and the MS spectra of FAME standards (Supelco 37 Component FAME Mix, ref. 47885-U; Sigma-Aldrich, St. Louis, MO, USA) and by MS spectrum comparison with chemical databases (Wiley 275 library, AOCS lipid library, and NIST 2014 Mass spectra library). The relative abundance (%) of each FA was calculated using the area of each peak obtained from integration with the Agilent MassHunter Qualitative10.0 software and considering the sum of all relative areas of the identified FAs.

### 2.4. Statistical Analyses

The relative FA composition per *S. pilchardus* from each landing location was used to assemble a resemblance matrix among samples using Euclidean distances, following a log (x + 1) transformation to emphasize differences among locations [23]. Only FAs with a relative abundance ≥ 1% of the total pool of FAs were considered for statistical analyses. The statistical differences (*p* < 0.05) in FA profiles among locations were tested using the vegan adonis() function for permutational multivariate analysis of variance (PERMANOVA) [23]. A Boruta analysis, using the “TentativeRoughFix” function, was performed to select the most relevant FAs to discriminate samples from different locations [24]. Subsequently, a linear discriminant analysis (LDA) was employed to evaluate the possibility of successfully discriminating the geographic origin of sampled specimens using these FAs present in the white muscle of *S. pilchardus*. All statistical analyses were performed in R [25].

### 2.5. Environmental Conditions: Upwelling Index and Sea Surface Temperature

As upwelling plays a key role on the shaping and condition of *S. pilchardus* populations occurring in the Iberian Atlantic coast [26], average upwelling indexes for each surveyed location were calculated using data from buoys operated by the Instituto Español de Oceanografía [27]. Four measurements were retrieved from each location per day (one every 6 h), for every day of the month. The results presented are averages of 30 days prior to the day of capture, presented in m^3^ s^−1^ km^−1^.

Sea surface temperature (SST) data were retrieved from the National Oceanic and Atmospheric Administration’s global temperature maps [28] and calculated as monthly temperature percentiles. These allowed for relative temperature comparisons between the northern (Cor, RP, VC, and Mat) and the southern (Pe, Ses, and Por) locations surveyed in the present study.

## 3. Results

### 3.1. Fatty Acid Composition

A total of 35 FAs were identified from the white muscle of *S. pilchardus*, some of which (e.g., 12:0, 17:1*n*-9, 18:1*n*-5, 22:5*n*-6) were exclusive to one or two of the surveyed sampling sites (Appendix A). Of these, the most abundant FA (relative abundance ≥ 1%) were selected for FA profiling and to investigate their potential use for the geographical discrimination of *S. pilchardus* samples collected at the seven locations surveyed along the Iberian Atlantic coast (Table 1). Docosahexaenoic acid (22:6*n*-3, DHA), eicosapentaenoic acid (20:5*n*-3, EPA), and palmitic acid (16:0) were the most abundant FAs in all samples, representing 20.9%, 15.9%, and 17.9% of the total FA pool, respectively (Appendix A). Polyunsaturated FAs (PUFAs) were the most abundant class across all locations (44.6% in RP to 52.4% in Ses of the total FA pool), followed by saturated FAs (SFAs) in all samples except Ses, where monounsaturated FAs (MUFAs) were the second most abundant FA class (Table 1). Within PUFAs, DHA was the most abundant in all samples (from 20.9% in Por to 24.6% in Cor), except Pe, where EPA was the most abundant (22.2%). Oleic acid (18:1*n*-9) was the most abundant MUFA (from 7.1% in RP to 9.7% in Por) and 16:0 the most abundant SFA in all samples (from 14.2% in Ses to 23.3% in RP). In RP, 16:0 was the most abundant FA, surpassing the relative abundance of DHA. The EPA/DHA ratio varied largely among *S. pilchardus* samples, with a 2-fold increase from Cor (0.56) to Pe (1.22). The differences in FA profiles led to significant differences between all *S. pilchardus* landed in the fishing harbors surveyed, as revealed by PERMANOVA results (*p* < 0.05, Table 2).

### 3.2. Fatty Acids as Markers of Geographic Origin

According to the Boruta analysis performed, the most important FAs to successfully discriminate between the seven landing locations were (in increasing order of relevance) myristic acid (14:0), DHA (22:6*n*-3), docosapentaenoic acid (DPA, 22:5*n*-3), stearic acid (18:0), EPA (20:5*n*-3), palmitoleic acid (16:1*n*-7), palmitic acid (16:0), and vaccenic acid (18:1*n*-7) (Figure 2).

The relative abundances of these FAs were generally higher in samples from RP than in samples from Cor and Pe, except for 18:1*n*-7, 20:5*n*-3, and 22:5*n*-3, which were higher in Pe than in RP and Cor, and 22:6*n*-3, which was higher in Cor than in RP and Pe (Table 1). These differences contributed to the contrasting levels of total SFAs, MUFAs, and PUFAs in *S. pilchardus* from Cor, RP, and Pe (Table 1). Whereas Pe and RP displayed almost twice the relative abundance of total PUFAs than total MUFAs, differences between the relative abundances of the three FA classes were not that pronounced in Cor. *Sardina pilchardus* from Ses displayed the highest relative abundance of PUFAs and almost identical proportions of SFAs and MUFAs, in which they differed from *S. pilchardus* captured in Pe, RP, and Cor. These differences are reflected in the LDA plot (Figure 3), where samples from RP, Cor, and Pe appeared well separated from those collected in VC, Mat, Ses, and Por, which were overlapped. While the first discriminant function explained 20.6% of the variation recorded in the white muscle FA profiles, mostly associated with the relative abundance of 22:6*n*-3, the second discriminant function explained 66.3% of that variation, mostly associated with the relative abundances of 16:0 and 22:6*n*-3.

Results from the LDA were further supported by the percentages of correct classification (Table 3). It is worth highlighting that RP and Pe samples, which appeared well separated in the LDA plot, showed the highest percentage of correct classification (100%), while VC and Mat, overlapping in the LDA plot, showed the highest misclassifications (40% and 20%, respectively, with a correct classification of 60% and 80%, respectively). Accordingly, four VC specimens were erroneously allocated to Cor, Mat, Ses, and Por, and two Mat specimens were erroneously allocated to VC. In *S. pilchardus* samples landed at Cor, Ses, and Por, a single specimen was misclassified (correct classification 90% in both locations).

### 3.3. Upwelling Index and Sea Surface Temperature

The upwelling index and SST percentile for each location are summarized in Table 4. Cor and RP, the northernmost locations (Figure 1), presented the highest upwelling values and ranked in the highest SST percentile. The lowest upwelling value was recorded in Ses, which ranked in the lower SST percentile.

## 4. Discussion

While, for the first time ever, the aquaculture volume of aquatic animals surpassed that of fisheries (accounting for 51% and 49%, respectively), marine finfish captures accounted for almost 70 million tons in 2022, representing 95% of the total volume of marine finfish production worldwide [29]. Consequently, there is an increased concern with the sustainability of these fisheries, but also with the quality, safety, and origin of consumed fish. These concerns have intensified in last few decades due to globalization and disease outbreaks related to food consumption (e.g., mad cow disease, avian flu, COVID-19) [14,30]), including bacterial or viral infections driven by the consumption of raw and/or undercooked fish [31]. Traceability is therefore paramount for ensuring the safety of consumed fish, tracking sustainable fishing, and combatting IUU fishing [32]. As the FA composition of a marine organism reflects its diet and environment (FAs are linked to specific basal components of the food web), as well as its metabolism (certain essential FAs must be acquired from the diet and are selectively retained in the muscle), the profiling of FAs is particularly effective when one wants to confirm the geographical origin of seafood [10,13,33]. In the present study, all sampled *S. pilchardus* belonged to the same metapopulation [34]. Therefore, differences in metabolism produced from genetic differences were not expected (a priori), and differences in FA profiles are likely due to contrasting environmental conditions and/or dietary regimes that are experienced by these fish in different fishing areas. As previously referred, FA profiling was selected in the present study as the best approach to confirm the origin of *S. pilchardus* landed in seven fishing ports along the Iberian Atlantic coast, from A Coruña in the north of Spain to Portimão in the South of Portugal.

Overall, *S. pilchardus* from the seven locations had a higher proportion of total PUFAs, particularly *n*-3 PUFAs, than total SFAs or total MUFAs, in line with that which had already been reported for this species in the areas where it occurs, including the Iberian Atlantic coast [35,36]. This high proportion of PUFAs is likely promoted by the consumption of phytoplanktonic green algae and cryptophytes (single-cell biflagellate algae), as nearly 60% of their total FAs are PUFAs, while SFAs and MUFAs are the prevalent FAs in dinoflagellates and blue-green algae, respectively; diatoms and haptophytes have similar proportions of these three FA classes [37].

The two most abundant FAs were, in general, DHA (22:6*n*-3) and palmitic acid (16:0), in line with the FA profiles previously reported for *S. pilchardus* [35,36,38]. However, EPA (20:5*n*-3) was the most abundant FA in samples from Pe and the second most abundant in samples originating from Ses, with the proportions of all three of these FAs varying among samples from different locations. The EPA/DHA ratios reflected the relative importance of these FAs in the seven locations, with these values suggesting that *S. pilchardus* from Pe and Ses feed more on green algae and diatoms, while *S. pilchardus* from Cor and Mat feed preferentially on dinoflagellates and haptophytes, as the former phytoplanktonic groups are richer in EPA and the latter in DHA [37]. Regarding SFAs, those with 16 carbons are the signature of silica-rich diatoms [37]. Therefore, the presence of 16:0 at a higher relative abundance in the white muscle of *S. pilchardus* from all locations suggests that diatoms are an important part of this planktivorous fish diet along the Atlantic Iberian coast.

The contrasting FA compositions recorded suggest that, although geographically close, *S. pilchardus* from RP appear to consume more diatoms and cryptophytes than those from Cor, thereby presenting higher levels of diatom-associated FA markers, namely, palmitoleic acid (16:1*n*-7) and EPA, whereas S*. pilchardus* from Cor consume more phytoplanktonic green algae and dinoflagellates, thereby presenting higher levels of dinoflagellate-associated markers, namely, stearidonic acid (18:4*n*-3) and DHA [39]. This is in agreement with diatoms being dominant throughout the year in Atlantic Galician Rías [40] and cryptophytes preferentially inhabiting still and non-eutrophic waters [41], as is the case of RP, whereas phytoplanktonic green algae and dinoflagellates are responsible for most oceanic primary production [37,42].

*Sardina pilchardus* from Pe and Ses presented much higher relative abundances of 6,9,12,15-hexadecatetraenoic acid (16:4*n*-1) and EPA when compared to those from the other fishing harbors, supporting that they feed on diatoms. Specimens from VC, Mat, and Por appeared clustered in the center of the LDA plot, as their FA profiles were generally similar. Exceptions were the high relative abundance of oleic acid (18:1*n*-9) in *S. pilchardus* from Por (the highest among locations) and DHA in those from Mat (the second highest among locations).

For what concerns the environmental conditions, SST was generally warmer than average in all locations, but the upwelling indexes of Cor and RP, which were similar, were two to eight times higher than those in the other five locations. An obvious consequence of upwelling is an increased concentration of nutrients that originate from deeper water layers, followed by phytoplankton blooms with nutrient-rich biomass [43]. Although presenting the lowest upwelling index, this does not seem to have affected the FA profile of *S. pilchardus* from Ses, as their FA profiles were similar to that of conspecifics landed in locations displaying higher upwelling intensity along the Portuguese coast (i.e., VC, Mat, and Por).

## 5. Conclusions

The present study highlights the discrimination potential of FA profiles in the white muscle of *S. pilchardus*. Indeed, individual FAs and FA classes present in this biological matrix allowed for the confirmation of the geographic origin of specimens landed in different commercial fishing harbors, located at different spatial distances from each other, thus allowing traceability guidelines currently in place to be enforced by national and EU authorities on fishery products. By using only eight of the 35 FAs identified in the white muscle of *S. pilchardus*, an average correct allocation of 80% on landing locations was achieved. Future studies are needed to refine this approach and investigate if and how seasonal and/or interannual variability in the FA profile of *S. pilchardus* can either fade or enhance the resolution already achieved with these natural biochemical fingerprints. The accurate identification of the geographic origin of landed *S. pilchardus* is paramount to successfully managing the Iberian stock of this commercially and ecologically important species, as overexploitation and IUU fishing practices (e.g., mislabeling of fishing area to dodge landing quotas) may continue to jeopardize the sustainability of this socially and economically important fishery.

## Figures and Tables

**Figure 1 foods-14-00120-f001:**
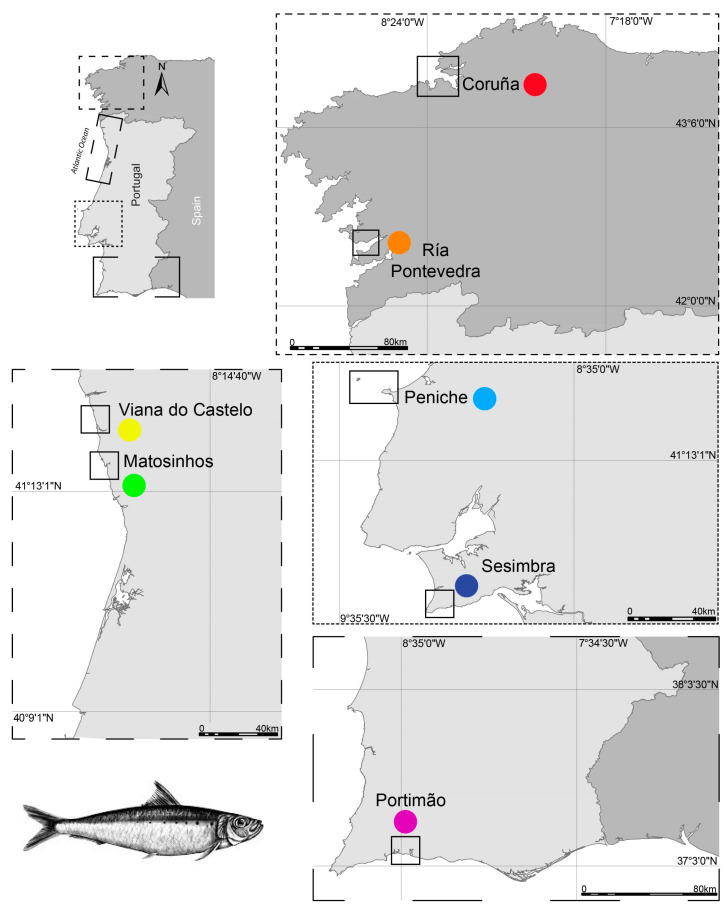
Landing locations of *Sardina pilchardus* along the Iberian Atlantic coast surveyed in the present study: Coruña (Cor, 43°19′25.1″ N 8°48′29.3″ W), Ría de Pontevedra (RP, 42°19′39.0″ N 8°47′06.9″ W), Viana do Castelo (VC, 41°41′09.1″ N 8°50′14.0″ W), Matosinhos (Mat, 41°10′59.8″ N 8°41′52.2″ W), Peniche (Pe, 39°21′20.6″ N 9°22′19.8″ W), Sesimbra (Ses, 38°26′24.7″ N 9°06′45.7″ W), and Portimão (Por, 37°07′58.6″ N 8°31′33.6″ W).

**Figure 2 foods-14-00120-f002:**
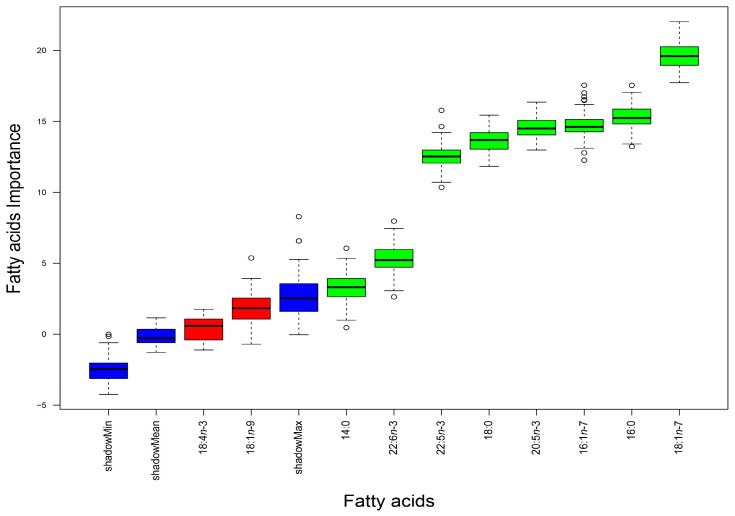
Box plots of Z-scores obtained by the Boruta algorithm to determine which fatty acids (FAs) in the white muscle of *Sardina pilchardus* are the most significant to successfully discriminate individuals landed at different locations along the Iberian Atlantic coast. As the Boruta algorithm first duplicates the dataset, shuffles these values (termed shadow features), and finds their importance (using a random forest classifier) to then compare the importance of the real data with that of its shadow, the FAs with higher Z-scores than the maximum Z-scores of their shadow features (shadowMax) are considered significant for the discrimination. Green box plots indicate the significant FAs, and the higher their position in the plot, the higher their importance for discriminating *S. pilchardus* from the landing locations. Box plots in red indicate FAs that were not considered significant for the discrimination, as they did not overcome the shadowMax.

**Figure 3 foods-14-00120-f003:**
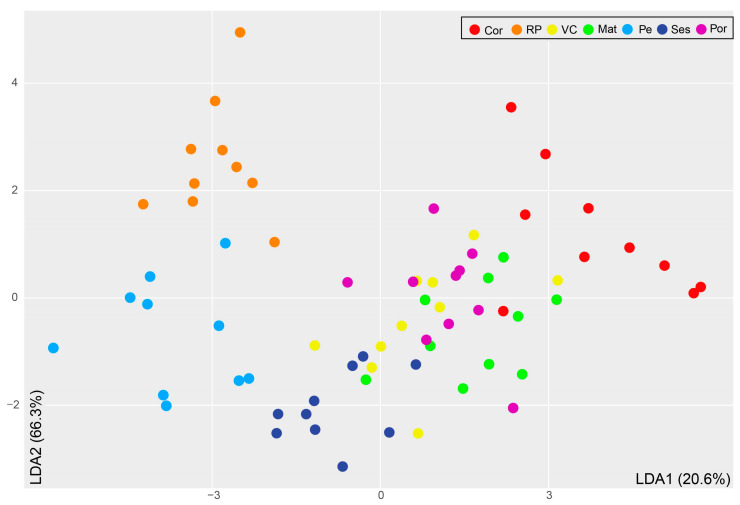
Linear discriminant analysis (LDA) of *Sardina pilchardus* collected from seven locations along the Iberian Atlantic coast based on the fatty acid profile of their white muscle, considered most significant by the Boruta algorithm to discriminate them. The plot evidences the distribution of the individuals from each location (n = 10) in the space defined by the first and second discriminant functions (LDA1 and LDA2, respectively), which explained 20.6% and 66.3% of the observed variance, respectively. Abbreviations: Cor: Coruña; RP: Ría de Pontevedra; VC: Viana do Castelo; Mat: Matosinhos; Pe: Peniche; Ses: Sesimbra; Por: Portimão.

**Table 1 foods-14-00120-t001:** Average fatty acid (FA) composition (relative abundance of total pool of FA (%) ± SD; only the most abundant FAs are presented, relative abundance ≥ 1%) of *Sardina pilchardus* white muscle, sampled from individuals obtained in the seven landing locations along the Iberian Atlantic coast (n = 10 specimens per location). Total FA composition is presented in Appendix A.

FA	Cor	RP	VC	Mat	Pe	Ses	Por
14:0	3.57 ± 0.43	4.08 ± 1.07	3.83 ± 0.34	3.76 ± 0.79	3.44 ± 0.30	3.43 ± 0.40	3.37 ± 0.33
16:0	19.54 ± 2.70	23.31 ± 2.69	17.41 ± 2.21	17.15 ± 1.98	17.02 ± 1.78	14.23 ± 0.90	17.89 ± 1.95
16:1*n*-7	4.38 ± 0.78	7.35 ± 1.62	5.96 ± 0.81	5.23 ± 0.80	6.40 ± 0.57	6.30 ± 0.52	5.30 ± 0.85
16:4*n*-1	0.49 ± 0.31	0.75 ± 0.63	0.62 ± 0.68	0.61 ± 0.23	1.72 ± 0.36	1.45 ± 0.32	0.76 ± 0.18
18:0	6.40 ± 1.84	9.96 ± 2.54	5.68 ± 0.84	5.51 ± 1.25	7.27 ± 1.22	4.81 ± 0.43	6.28 ± 1.04
18:1*n*-9	8.31 ± 2.29	7.17 ± 2.31	8.92 ± 1.61	7.53 ± 1.28	8.93 ± 0.92	7.86 ± 1.72	9.75 ± 2.31
18:1*n*-7	2.07 ± 0.14	3.01 ± 0.30	2.47 ± 0.22	2.33 ± 0.10	3.16 ± 0.21	2.75 ± 0.15	2.39 ± 0.24
18:4*n*-3	2.21 ± 0.38	1.87 ± 0.38	2.13 ± 0.33	2.13 ± 0.29	2.00 ± 0.20	2.07 ± 0.32	2.08 ± 0.26
20:5*n*-3	13.75 ± 3.12	19.39 ± 4.76	17.22 ± 3.48	15.80 ± 1.65	22.24 ± 1.66	20.56 ± 1.21	15.92 ± 1.67
22:5*n*-3	1.34 ± 0.29	1.34 ± 0.46	1.91 ± 0.30	1.90 ± 0.33	2.42 ± 0.38	2.12 ± 0.28	1.84 ± 0.33
22:6*n*-3	24.60 ± 1.63	21.03 ± 7.38	21.81 ± 3.10	24.25 ± 3.25	18.22 ± 3.09	21.30 ± 3.09	20.89 ± 2.36
EPA/DHA	0.56	0.92	0.79	0.65	1.22	0.97	0.76
∑SFA	29.95 ± 4.50	37.84 ± 4.51	27.27 ± 2.79	26.95 ± 2.85	28.26 ± 2.77	23.24 ± 1.42	28.34 ± 2.93
∑MUFA	24.05 ± 3.42	17.53 ± 3.74	26.46 ± 3.69	24.59 ± 2.59	20.37 ± 2.77	23.59 ± 1.93	25.77 ± 2.93
∑PUFA	45.75 ± 3.22	44.64 ± 4.98	46.12 ± 3.25	48.26 ± 3.27	51.02 ± 2.40	52.47 ± 2.23	45.65 ± 1.21
∑*n*-3	43.02 ± 2.59	43.64 ± 4.96	44.00 ± 2.75	45.05 ± 3.20	45.77 ± 2.10	47.01 ± 2.74	41.99 ± 1.48

Abbreviations: Cor: Coruña; RP: Ría de Pontevedra; VC: Viana do Castelo; Mat: Matosinhos; Pe: Peniche; Ses: Sesimbra; Por: Portimão. EPA: eicosapentaenoic acid (20:5*n*-3); DHA: docosahexaenoic acid (22:6*n*-3). SFA: saturated fatty acid; MUFA: monounsaturated fatty acid; PUFA: polyunsaturated fatty acid. ∑ refers to all FAs identified for that specific class; please refer to Appendix A.

**Table 2 foods-14-00120-t002:** Pair-wise PERMANOVA of *Sardina pilchardus* white muscle fatty acid profiles.

	Cor	RP	VC	Mat	Pe	Ses
**Cor**						
**RP**	*p* < 0.001					
**VC**	*p* = 0.002	*p* < 0.001				
**Mat**	*p* = 0.013	*p* < 0.001	*p* = 0.021			
**Pe**	*p* < 0.001	*p* < 0.001	*p* < 0.001	*p* < 0.001		
**Ses**	*p* < 0.001	*p* < 0.001	*p* = 0.002	*p* < 0.001	*p* < 0.001	
**Por**	*p* = 0.002	*p* < 0.001	*p* = 0.023	*p* = 0.007	*p* < 0.001	*p* < 0.001

Abbreviations: Cor: Coruña; RP: Ría de Pontevedra; VC: Viana do Castelo; Mat: Matosinhos; Pe: Peniche; Ses: Sesimbra; Por: Portimão.

**Table 3 foods-14-00120-t003:** Classification success (by landing location) of the linear discriminant analysis (LDA) based on *Sardina pilchardus* white muscle fatty acid profiles.

Original Location	Predicted Location							Correct Classification (%)
	Cor	RP	VC	Mat	Pe	Ses	Por	
**Cor**	90	0	0	10	0	0	0	90.0
**RP**	00	100	0	0	0	0	0	100.0
**VC**	10	0	60	10	0	10	10	60.0
**Mat**	0	0	20	80	0	0	0	80.0
**Pe**	0	0	0	0	100	0	0	100.0
**Ses**	0	0	0	0	0	90	10	90.0
**Por**	0	0	0	0	0	10	90	90.0
Average correct classification								82.9

Abbreviations: Cor: Coruña; RP: Ría de Pontevedra; VC: Viana do Castelo; Mat: Matosinhos; Pe: Peniche; Ses: Sesimbra; Por: Portimão.

**Table 4 foods-14-00120-t004:** Upwelling index and temperature percentile of the seven landing locations of *Sardina pilchardus* surveyed along the Iberian Atlantic coast.

	Cor	RP	VC	Mat	Pe	Ses	Por
Upwelling index (m^3^ s^−1^ km^−1^)	1712.2 (June)	1648.5 (June)	703.6 (July)	552.4 (July)	789.2 (July)	214.0 (July)	769.5 (July)
Temperature percentile (June)	Warmer than average	Warmer than average	Warmer than average	Warmer than average	Near average	Near average	Near average
Temperature percentile (July)	Much warmer than average	Much warmer than average	Much warmer than average	Much warmer than average	Warmer than average	Warmer than average	Warmer than average

Abbreviations: Cor: Coruña; RP: Ría de Pontevedra; VC: Viana do Castelo; Mat: Matosinhos; Pe: Peniche; Ses: Sesimbra; Por: Portimão.

## Data Availability

The original contributions presented in the study are included in the article/Appendix A, further inquiries can be directed to the corresponding authors.

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
