# Peer review of "Fatty Acids of European Sardine (Sardina pilchardus) White Muscle Can Discriminate Geographic Origin Along the Iberian Atlantic Coastâ€"

_foods, 2025, doi:10.3390/foods14010120_

Round 1
Reviewer 1 Report
Comments and Suggestions for Authors
In this manuscript, the authors tried to establish a method to discriminate geographic origins of European sardine based on the differences in fatty acids. The experiments were logically designed and the results were convincing. However, to discriminate geographic origins of fish, more convenient and specific method by genetic technology (PCR) has been well established. The present work, however, is time consuming and with high cost. Thus, I do not think it is a practical way to discriminate geographic origins of European sardine based on the differences in fatty acids. I recommend the authors to change their research focus to comparison of fatty acid composition in different geographic origins of European sardine and change the manuscript title accordingly. Only in this way, the manuscript maybe acceptable.
Author Response
Reviewer #1 Comments and Suggestions for Authors
Responses to each reviewer comment (RxCy) (x being the number of the reviewer and y the number of the comment by the same reviewer) are provided bellow as RxRy.
The anonymous reviewers are acknowledged, as their insightful comments and constructive criticism significantly helped to improve the overall quality of the final manuscript.
R1C1: In this manuscript, the authors tried to establish a method to discriminate geographic origins of European sardine based on the differences in fatty acids. The experiments were logically designed and the results were convincing.
R1R1: The authors acknowledge the positive feedback by Reviewer #1.
R1C2: However, to discriminate geographic origins of fish, more convenient and specific method by genetic technology (PCR) has been well established. The present work, however, is time consuming and with high cost. Thus, I do not think it is a practical way to discriminate geographic origins of European sardine based on the differences in fatty acids. I recommend the authors to change their research focus to comparison of fatty acid composition in different geographic origins of European sardine and change the manuscript title accordingly. Only in this way, the manuscript maybe acceptable.
R1R2: We respectfully disagree from Reviewer #1. While Reviewer #1 is certainly right when advocating a genetic approach using PCR technology if one aims to discriminate between different fishing stocks with a well-marked genetic structure that will make possible to flag differences between specimens originating from such stocks, this is not possible when working at smaller spatial scales, such as the ones employed in the present study. Indeed, in our study, all specimens sampled at different fishing grounds are indeed from the same fishing stock. Genetic connectivity significantly blurs the genetic signatures that could be used to confirm their geographic origin, while fatty acids reflect the abiotic conditions experienced by the fish and the influence of their diet. These two drivers play a key role in the shaping of the fatty acid profiles present in the biological tissues of European sardines. As such, fatty acid profiles, unlike genetic markers, can allow the discrimination of geographic origin of finfish and shellfish at a much smaller spatial scale (for some species, one may even discriminate their origin within the same estuary(!) and, even more easily, between adjacent estuaries). To the authors best knowledge, the most recent study on this topic is quite up to date: da Fonseca, R.R.; Campos, P.F.; Rey-Iglesia, A.; Barroso, G.V.; Bergeron, L.A.; Nande, M.; Tuya, F.; Abidli, S.; Pérez, M.; Riveiro, I.; et al. Population Genomics Reveals the Underlying Structure of the Small Pelagic European Sardine and Suggests Low Connectivity within Macaronesia. Genes 2024, 15, 170. https://doi.org/10.3390/genes15020170. The study by da Fonseca and colleagues refers that their findings support “…at least three genetic clusters. One includes individuals from Azores and Madeira, with evidence of substructure separating these two archipelagos in the Atlantic. Another cluster broadly corresponds to the center of the distribution, including the sampling sites around Iberia, separated by the Almeria–Oran front from the third cluster that includes all of the Mediterranean samples, except those from the Alboran Sea.”. Hence, all specimens surveyed in our study will be grouped within the “cluster” that broadly corresponds to “the center of the distribution” that includes all “sampling sites around Iberia”, which is precisely the location of our study: seven landing ports along the Iberian Atlantic coast. Overall, while we acknowledge in our manuscript that our approach steal needs to be refined, we reaffirm that our findings support the use of fatty acid profiles to verify claims of geographic origin for European Sardine being landed along the Iberian Atlantic coast. We would also like to remark that when specifically knowing which fatty acids contribute the most to discriminate being geographic origins, costs and processing time to determine fatty acid profiles for the purpose of seafood traceability can match, or even outcompete, those addressing employing genetic markers.
Reviewer 2 Report
Comments and Suggestions for Authors
The article entitled “Fatty acids of European sardine (Sardina pilchardus) white muscle can successfully discriminate its geographic origin along the Iberian Atlantic coast” addresses an interesting topic. Interesting results are presented, but the discussion needs to be considerably improved.
According to the results presented, it is not clear how it is possible to differentiate the origin of the sardine. The authors must work on this idea, as it is the essence of their research. They mention that the fatty acid 18:1n7 is the best biomarker, but taking into account the mean and its standard deviations, the localities Cor, VC, Mat, Ses and Por, present very similar results, the same happens with the localities RP and Pe. In this sense, I think that the conclusions are not entirely correct.
General comments
Line 35, 44, 175, 306, 323, 369: scientific names should be written in italics
Line 93: what size does category 4 correspond to?
Line 123: CH2Cl2.....instead of CH2Cl2
Line 124: sonication for 1 min. Describe the equipment and sonication conditions used.
Line 173: R?????......... What does this mean?
Author Response
Reviewer #2 Comments and Suggestions for Authors
Responses to each reviewer comment (RxCy) (x being the number of the reviewer and y the number of the comment by the same reviewer) are provided bellow as RxRy.
The anonymous reviewers are acknowledged, as their insightful comments and constructive criticism significantly helped to improve the overall quality of the final manuscript.
R2C1: The article entitled “Fatty acids of European sardine (Sardina pilchardus) white muscle can successfully discriminate its geographic origin along the Iberian Atlantic coast” addresses an interesting topic.
R2R1: The authors acknowledge the positive feedback and constructive criticism by Reviewer #2.
R2C2: Interesting results are presented, but the discussion needs to be considerably improved. According to the results presented, it is not clear how it is possible to differentiate the origin of the sardine. The authors must work on this idea, as it is the essence of their research. They mention that the fatty acid 18:1n7 is the best biomarker, but taking into account the mean and its standard deviations, the localities Cor, VC, Mat, Ses and Por, present very similar results, the same happens with the localities RP and Pe.
R2R2: We do understand the concerns expressed by Reviewer #2 if one considers a single fatty acid (FA) individually (such as 18:1n7) to infer on significant differences and looking at the means and standard deviations recorded. However, we must highlight that we are looking at the whole pool of fatty acids of our biological matrix (the white muscle of S. pilchardus), to be more precise, to all FAs whose relative abundance accounts for ≥ 1% of the total pool of FAs. The PERMANOVA performed uses all these variables to account for significant differences in FA profiles, rather comparing FA individually for specimens landed in different fishing harbors. Moreover, when it comes to our linear discriminant analysis (LDA), it considers a set of FAs used as discriminant features, which the Boruta algorithm identifies. This algorithm relies on variable importance values that it repeatedly calculates, by running Random Forest models on our dataset for a predefined set of predictor variables. It creates what it is called a set of random noise variables associated with each feature as it shuffles each value's row location. This allows to establish a rank based on the permutation feature importance, as the algorithm detects if the associated error increases or decreases when the values of a feature are permuted. If such permuting values promote a change in the error, it means the feature is important for our model. In the case of our analysis, that rank featured the FA 18:1n7 as the most important, but that importance cannot be evaluated individually for that specific FA without taking in account the role that is also played by the other 7 FAs that were also highlighted by the algorithm (14:0, 16:0, 16:1n-7, 18:0, 20:5n-3, 22:5n-3 and 22:6n-3). What that higher position in the rank stands for is that if one does not consider FA 18:1n7 in our predictive model the error of that model increases much more. We hope that with these explanations on how the different statistical analysis performed in the present study operate, the doubts of Reviewer #2 on the validity of our findings and the way we discussed them in our manuscript are now clarified. If the Editor considers that any of these clarifications should be featured in our revised manuscript, the authors will certainly include them.
R2C3: In this sense, I think that the conclusions are not entirely correct.
R2R3: Please refer to our reply to R2C2. We hope that with the clarifications provided in R2C2 Reviewer #2 now find our conclusions to be supported by our findings.
General comments
R2C4: Line 35, 44, 175, 306, 323, 369: scientific names should be written in italics
R2C4: Thank you very much for spotting this issue. The manuscript was thoroughly reviewed to make sure that on its revised form all scientific names are now written in italics.
R2C5: Line 93: what size does category 4 correspond to?
R2R5: Indeed, this information was missing. For clarification the following information was added to the revised manuscript that now reads as follows: (T4 category, weighting between 15 and 28 g) (please refer to https://eur-lex.europa.eu/legal-content/EN/TXT/PDF/?uri=CELEX:31996R2406 pag. 11, Annex II Size Categories).
R2C6: Line 123: CH2Cl2.....instead of CH2Cl2
R2R6: Corrected as suggested.
R2C7: Line 124: sonication for 1 min. Describe the equipment and sonication conditions used.
R2R7: The description of the sonication procedure has now been detailed in our revised manuscript. It now reads: “sonication for 1 min using an automatic ultrasonic frequency of 35 kHz (Bandelin, Sonorex, RK 100)”
R2C8: Line 173: R?????......... What does this mean?
R2R8: R is a language and environment for statistical computing and graphics. Please refer to https://www.r-project.org/about.html for further information.
Reviewer 3 Report
Comments and Suggestions for Authors
The manuscript aims to reconstruct the traceability of fish products through geographical origin. This can be very useful for the control of sustainable and legal fishing.
The work is very interesting and well done, although there are aspects to be improved.
There is talk of "seven 93 relevant fishing harbors along the Iberian Atlantic coast" (line 93-94). According to which criteria? Geographical location, capacity, economic importance.... Please explain
The PERMANOVA which variables includes? Please specify in statistical analysis section.
How were upwelling and temperature included in the evaluation of FA? Were possible correlations or interactions assessed?
In Figure 2 legend the authors explain method about Boruta algorithm (lines 231-235). Perhaps these details are too methodological for a figure legend. Please moves this lines in statistical analysis
Please, the authors consider changing the title as: "Fatty acids of European sardine (Sardina pilchardus) white muscle to discriminate its geographic origin". The ability to discriminate is already a success, so ‘succesfully’ should not be included in the title; perhaps I would also eliminate the location of the study, but the authors can evaluate the suggestion.
Author Response
Reviewer #3 Comments and Suggestions for Authors
Responses to each reviewer comment (RxCy) (x being the number of the reviewer and y the number of the comment by the same reviewer) are provided bellow as RxRy.
The anonymous reviewers are acknowledged, as their insightful comments and constructive criticism significantly helped to improve the overall quality of the final manuscript.
R3C1: The manuscript aims to reconstruct the traceability of fish products through geographical origin. This can be very useful for the control of sustainable and legal fishing. The work is very interesting and well done, although there are aspects to be improved.
R3R1: The authors acknowledge the positive feedback and constructive criticism by Reviewer #3.
R3C2: There is talk of "seven relevant fishing harbors along the Iberian Atlantic coast" (line 93-94). According to which criteria? Geographical location, capacity, economic importance.... Please explain
R3R2: We agree with Reviewer #3 that our wording was not the best to refer to the selection of the different fishing harbors that were surveyed. For clarity, we have reworded this part of the text and it now reads as follows: “Landed S. pilchardus of similar commercial size (T4 category, weighting between 15 and 28 g) were obtained in seven fishing harbors regularly landing this species along the Iberian Atlantic coast during early summer…”.
R3C3: The PERMANOVA which variables includes? Please specify in statistical analysis section.
R3R3: The information requested by Reviewer #3 is already referred in subsection 2.4. Statistical analyses, it reads as follows: “Only FAs with a relative abundance ≥ 1% of the total pool of FAs were considered for statistical analyses.”. Hence, all FAs whose abundance ≥ 1% of the total pool of FAs were considered for each specimen of sardine (here plotted as our samples and their fatty acids as the variables monitored); Fishing Harbors were considered as Factor and each Fishing Harbor was considered a Level within the Factor, with these being compared pair-wise. Currently, we consider that PERMANOVA is a fairly well-established and well-known multivariate statistical method and, as such, further details, beyond those already provided in the manuscript, may not be required. However, if the Editor understands otherwise, we can certainly elaborate on the description of our PERMANOVA.
R3C4: How were upwelling and temperature included in the evaluation of FA? Were possible correlations or interactions assessed?
R3R4: This is another pertinent question raised by Reviewer #3. We have not investigated, in statistical terms, how upwelling indexes and sea surface temperatures could correlate or interact with the fatty acid profiles of the specimens that were sampled, as this would certainly go beyond the scope of the present publication and would require detailed time series to cover the whole lifespan of S. pilchardus surveyed in our study. These values were retrieved solely to help better understanding how contrasting are the oceanographic conditions over the different sampling locations and provide some rough guidelines to the authors when discussing the fatty acid profiles recorded, particularly those fatty acids that are known to be reliable trophic markers and/or more significantly influenced by water temperature. Sardina pilchardus primarily feeds on planktonic crustaceans whose fatty acid composition is influenced by oceanographic conditions, such as temperature, or nutrient availability. These conditions play a crucial role in shaping the nutritional quality of the plankton, ultimately impacting the energy transfer within the marine food web. All this interpretation was solely supported by existing scientific literature on the topic and not by any statistical analysis performed by the authors (as the scope of our study was on investigating the potential existence of variability on fatty acid profiles of S. pilchardus white muscle and if this could be used to reliably trace its geographic origin, rather than shedding light on why fatty acid profiles vary between the different sampling locations surveyed; nonetheless, this is a topic of scientific interest and certainly worth investigating! As the authors are making their data freely available as supplementary material, researchers interested in this topic may certainly explore potential correlations and/or interactions between oceanographic conditions and the fatty acid profiles of S. pilchardus white muscle).
R3C5: In Figure 2 legend the authors explain method about Boruta algorithm (lines 231-235). Perhaps these details are too methodological for a figure legend. Please moves this lines in statistical analysis.
R3R5: We acknowledge the suggestion by Reviewer #3, as it is not common to explain a method in the caption of a figure detailing its results. However, not all readers are familiar with the Boruta algorithm and given the key role it plays on the selection of the features (fatty acids) being employed to run the predictive model employed (LDA) we consider that it will be more informative and easier to follow to the reader if these details are provided in the caption. Please note that we have already used this approach in previous publications (e.g., Mamede, R.; Domingues, M.R.; Ferreira da Silva, E.; Patinha, C.; Calado, R. Combined Use of Fatty Acid Profiles and Elemental Fingerprints to Trace the Geographic Origin of Live Baits for Sports Fishing: The Solitary TubeWorm (Diopatra neapolitana, Annelida, Onuphidae) as a Case Study. Animals 2024, 14, 1361. https://doi.org/10.3390/ani14091361 and Mamede, R.; Santos, A.; Díaz, S.; Ferreira da Silva, E.; Patinha, C.; Calado, R.; Ricardo, F. Elemental fingerprints of bivalve shells (Ruditapes decussatus and R. philippinarum) as natural tags to confirm their geographic origin and expose fraudulent trade practices. Food Control 2022, 135, 108785, https://doi.org/10.1016/j.foodcont.2021.108785). As such, unless the Editor decides otherwise, we would respectfully ask to maintain the caption text as it is.
R3C6: Please, the authors consider changing the title as: "Fatty acids of European sardine (Sardina pilchardus) white muscle to discriminate its geographic origin". The ability to discriminate is already a success, so ‘succesfully’ should not be included in the title; perhaps I would also eliminate the location of the study, but the authors can evaluate the suggestion.
R3R6: This is a very good point put forward by Reviewer #3. As suggested, to avoid redundancy of wording in the title we have revised it. Our revised title now reads: “Fatty acids of European sardine (Sardina pilchardus) white muscle can discriminate geographic origin along the Iberian Atlantic coast”.
Reviewer 4 Report
Comments and Suggestions for Authors
Calado and colleagues conducted a research by sampling, GC-MS measuring and machine learing. The storytelling is good, but the conclusion was not solid.
1. The number of sampling indivisuals was not enough to conduct a feature selection. Please clarify.
2. You only use ONE method to prove your deduction, obviously, its not solid. For you reference-https://doi.org/10.3390/foods9111615
Author Response
Reviewer #4 Comments and Suggestions for Authors
Responses to each reviewer comment (RxCy) (x being the number of the reviewer and y the number of the comment by the same reviewer) are provided bellow as RxRy.
The anonymous reviewers are acknowledged, as their insightful comments and constructive criticism significantly helped to improve the overall quality of the final manuscript.
R4C1: Calado and colleagues conducted a research by sampling, GC-MS measuring and machine learning. The storytelling is good, but the conclusion was not solid.
- The number of sampling individuals was not enough to conduct a feature selection. Please clarify.
R4R1: The authors acknowledge the positive feedback by Reviewer #4 and perfectly understand the concerns expressed on the number of specimens screened per location (n = 10 individuals). One must acknowledge that, as “a rule of thumb” it is recommended that at least 10 data points are need for each model parameter employed when doing this type of studies. Moreover, when one performs this type of analysis, feature size should not exceed N−1 (where N is sample size). We check these two “assumptions” that methodologically legitimate our approach by having the minimum number of data points per model (10, which equals the 10 specimens sampled per location) and employing 8 fatty acids (previously selected by the Boruta algorithm as the most significant to successfully discriminate individuals), thus having a feature size of 8, which is inferior to the maximum that would be allowed in our experimental design (as our sample size (N) = 10, our maximum feature = 10-1= 9). We highlight that we do understand the concerns expressed by Reviewer #4 and that we took the required actions to analyze the minimum of specimens and the minimum number of features possible to speed-up analysis without compromising the reliability of our statistical analysis.
R4C2: 2. You only use ONE method to prove your deduction, obviously, its not solid. For you reference-https://doi.org/10.3390/foods9111615
R4R2: Again, Reviewer #4 raises a key-point in our approach: why and how have we selected a single method (over several others that are also valid choices) to run the present analysis? We would like to bring the reviewer’s attention that the present study builds upon our previous publications on seafood traceability using fatty acids as predictive features:
Ricardo, F.; Pimentel, T.; Maciel, E.; Moreira, A.S.P.; Rosário Domingues, M.; Calado, R. Fatty acid dynamics of the adductor muscle of live cockles (Cerastoderma edule) during their shelf-life and its relevance for traceability of geographic origin. Food Control 2017, 77, 192-198, https://doi.org/10.1016/j.foodcont.2017.01.012
Ricardo, F.; Maciel, E.; Domingues, M.R.; Calado, R. Spatio-temporal variability in the fatty acid profile of the adductor muscle of the common cockle Cerastoderma edule and its relevance for tracing geographic origin. Food Control 2017, 81, 173-180, https://doi.org/10.1016/j.foodcont.2017.06.005
It is true that we have also successfully employed other predictive model approaches, but the goal of our manuscript was not to determine which of the many predictive models that exist can achieve the best prediction performance, but rather to look at the problem a few steps back and start by validating the white muscle of S. pilchardus as a reliable biological matrix to trace the geographic origin of this pelagic fish and that fatty acids could indeed be used as good predictive features. In these two previous publications we had also worked with a total of only 10 specimens per sampling location/time point and managed to successfully discriminate them with acceptable classification accuracies. Hence, we have decided to follow this rather conservative approach and use LDA was to evaluate the possibility of successfully discriminating the geographic origin of sampled specimens using these FA present in the white muscle of S. pilchardus.
It is also worth referring that in case LDA was not the best predictive model and some other new model is suggested through a Machine Learning-Based Analysis, a higher classification accuracy would be achieved and never a lower one than that that we present in our current manuscript.
As we currently gather more data from a new ongoing sampling, that covers a wider geographic area, different time points and samples more specimens per location/time point (n=30), will we use a similar approach to the one described by the reviewer in the publication that was gently recommended (Fu, B.; Kaneko, G.; Xie, J.; Li, Z.; Tian, J.; Gong, W.; Zhang, K.; Xia, Y.; Yu, E.; Wang, G. Value-Added Carp Products: Multi-Class Evaluation of Crisp Grass Carp by Machine Learning-Based Analysis of Blood Indexes. Foods 2020, 9, 1615. https://doi.org/10.3390/foods9111615) to either confirm LDA as the best predictive model or points towards other viable approaches that can make possible to achieve higher classification accuracies. We sincerely thank the reviewer for this important recommendation that will allow us to refine our predictive approach in future studies.
Round 2
Reviewer 1 Report
Comments and Suggestions for Authors
It is recommennded that the authors to analyze several typical fat acids biosynthesis genes in sardine from different geographic origins and compare their identities. Based on this genetic comparison results, the manuscript will be more convincing.
Author Response
Reviewer #1 Comments and Suggestions for Authors
Responses to each reviewer comment (RxCy) (x being the number of the reviewer and y the number of the comment by the same reviewer) are provided bellow as RxRy.
The anonymous reviewers are acknowledged, as their insightful comments and constructive criticism significantly helped to improve the overall quality of the final manuscript.
R1C1: It is recommended that the authors to analyze several typical fat acids biosynthesis genes in sardine from different geographic origins and compare their identities. Based on this genetic comparison results, the manuscript will be more convincing.
R1R1: The authors acknowledge this insightful suggestion by Reviewer #1, but such approach would go far beyond the scope of the present study. To put forward a study “to analyze several typical fat acids biosynthesis genes in sardine from different geographic origins and compare their identities” as suggested by Reviewer #1, a whole new research project would have to be started and funded (and this will not be an inexpensive project to fund)! As such, this is not a feasible recommendation to improve the present manuscript. Moreover, we have clearly detailed in our reply R1R2 in the previous round of peer review:
“To the authors best knowledge, the most recent study on this topic is quite up to date: da Fonseca, R.R.; Campos, P.F.; Rey-Iglesia, A.; Barroso, G.V.; Bergeron, L.A.; Nande, M.; Tuya, F.; Abidli, S.; Pérez, M.; Riveiro, I.; et al. Population Genomics Reveals the Underlying Structure of the Small Pelagic European Sardine and Suggests Low Connectivity within Macaronesia. Genes 2024, 15, 170. https://doi.org/10.3390/genes15020170. The study by da Fonseca and colleagues refers that their findings support “…at least three genetic clusters. One includes individuals from Azores and Madeira, with evidence of substructure separating these two archipelagos in the Atlantic. Another cluster broadly corresponds to the center of the distribution, including the sampling sites around Iberia, separated by the Almeria–Oran front from the third cluster that includes all of the Mediterranean samples, except those from the Alboran Sea.”. Hence, all specimens surveyed in our study will be grouped within the “cluster” that broadly corresponds to “the center of the distribution” that includes all “sampling sites around Iberia”, which is precisely the location of our study: seven landing ports along the Iberian Atlantic coast. “
As such, it is highly likely that an expensive approach as the one suggested by Reviewer #1 may end-up revealing that sardine genes involved in fatty acids biosynthesis ado not differ between the different populations sampled, as they are all part of the “cluster” that broadly corresponds to “the center of the distribution” that includes all “sampling sites around Iberia”. Moreover, the fatty acids that commonly allow to discriminate between different populations of marine organisms from the same species are those termed as “trophic markers”. These fatty acids result from the bioaccumulation of these biomolecules and are derived from dietary items, not from de novo synthesis by the organisms. In other words, the fatty acids we report in our work as being the most relevant ones to discriminate between sardines from different locations are those known to be fatty acid trophic markers, hence not being biosynthesised by the sardines. As such, investigating sardine genes involved in fatty acids biosynthesis can a priori be considered of little use when aiming to discriminate the geographic origin of motile marine organisms at small spatial scales.
Reviewer 2 Report
Comments and Suggestions for Authors
The authors did not take into consideration the following comments:
Line 177: R?????......... What does this mean?
According to the results presented, it is not clear how it is possible to differentiate the origin of the sardine. The authors must work on this idea, as it is the essence of their research. They mention that the fatty acid 18:1n7 is the best biomarker, but taking into account the mean and its standard deviations, the localities Cor, VC, Mat, Ses and Por, present very similar results, the same happens with the localities RP and Pe. In this sense, I think that the conclusions are not entirely correct.
Author Response
Reviewer #2 Comments and Suggestions for Authors
Responses to each reviewer comment (RxCy) (x being the number of the reviewer and y the number of the comment by the same reviewer) are provided bellow as RxRy.
The anonymous reviewers are acknowledged, as their insightful comments and constructive criticism significantly helped to improve the overall quality of the final manuscript.
R2C1:. The authors did not take into consideration the following comments: Line 177: R?????......... What does this mean?
R2R1: We would like to bring Reviewer #2 attention to our previous reply R2R8, where we have indeed replied to this comment. It read as follows:
“R is a language and environment for statistical computing and graphics. Please refer to https://www.r-project.org/about.html for further information.”.
R2C2: The authors did not take into consideration the following comments: According to the results presented, it is not clear how it is possible to differentiate the origin of the sardine. The authors must work on this idea, as it is the essence of their research. They mention that the fatty acid 18:1n7 is the best biomarker, but taking into account the mean and its standard deviations, the localities Cor, VC, Mat, Ses and Por, present very similar results, the same happens with the localities RP and Pe. In this sense, I think that the conclusions are not entirely correct.
R2R2: We would like to bring Reviewer #2 attention to our previous reply R2R2, where we have indeed replied to this comment. It read as follows:
“We do understand the concerns expressed by Reviewer #2 if one considers a single fatty acid (FA) individually (such as 18:1n7) to infer on significant differences and looking at the means and standard deviations recorded. However, we must highlight that we are looking at the whole pool of fatty acids of our biological matrix (the white muscle of S. pilchardus), to be more precise, to all FAs whose relative abundance accounts for ≥ 1% of the total pool of FAs. The PERMANOVA performed uses all these variables to account for significant differences in FA profiles, rather comparing FA individually for specimens landed in different fishing harbors. Moreover, when it comes to our linear discriminant analysis (LDA), it considers a set of FAs used as discriminant features, which the Boruta algorithm identifies. This algorithm relies on variable importance values that it repeatedly calculates, by running Random Forest models on our dataset for a predefined set of predictor variables. It creates what it is called a set of random noise variables associated with each feature as it shuffles each value's row location. This allows to establish a rank based on the permutation feature importance, as the algorithm detects if the associated error increases or decreases when the values of a feature are permuted. If such permuting values promote a change in the error, it means the feature is important for our model. In the case of our analysis, that rank featured the FA 18:1n7 as the most important, but that importance cannot be evaluated individually for that specific FA without taking in account the role that is also played by the other 7 FAs that were also highlighted by the algorithm (14:0, 16:0, 16:1n-7, 18:0, 20:5n-3, 22:5n-3 and 22:6n-3). What that higher position in the rank stands for is that if one does not consider FA 18:1n7 in our predictive model the error of that model increases much more. We hope that with these explanations on how the different statistical analysis performed in the present study operate, the doubts of Reviewer #2 on the validity of our findings and the way we discussed them in our manuscript are now clarified. If the Editor considers that any of these clarifications should be featured in our revised manuscript, the authors will certainly include them.”
Reviewer 4 Report
Comments and Suggestions for Authors
The authors carefully responded to my doubts and concerns. Also, the authors decreasing the copy rate in the present manuscript.
Author Response
Reviewer #4 Comments and Suggestions for Authors
Responses to each reviewer comment (RxCy) (x being the number of the reviewer and y the number of the comment by the same reviewer) are provided bellow as RxRy.
The anonymous reviewers are acknowledged, as their insightful comments and constructive criticism significantly helped to improve the overall quality of the final manuscript.
R4C1: The authors carefully responded to my doubts and concerns. Also, the authors decreasing the copy rate in the present manuscript.
R4R1: The authors acknowledge the positive feedback by Reviewer #4.